# Preliminary Evaluation of Blending, Tuning, and Scaling Parameters in ssGBLUP for Genomic Prediction Accuracy in South African Holstein Cattle

**DOI:** 10.3390/ani15192866

**Published:** 2025-09-30

**Authors:** Kgaogelo Stimela Mafolo, Michael D. MacNeil, Frederick W. C. Neser, Mahlako Linah Makgahlela

**Affiliations:** 1Agricultural Research Council, Animal Production, Private Bag X2, Irene 0062, South Africammakgahlela@arc.agric.za (M.L.M.); 2University of the Free State, Department of Animal Science, P.O. Box 339, Bloemfontein 9301, South Africa; neserfw@ufs.ac.za; 3Delta G, Miles City, MT 59301, USA

**Keywords:** genomic prediction, ssGBLUP, blending, tuning, scaling, Holstein, accuracy

## Abstract

Predicting the genetic merit of dairy cattle animals is critical for increasing milk production and breeding efficiency. This study looked at how different adjustments to a genomic prediction method known as single-step genomic best linear unbiased prediction (ssGBLUP) affected the accuracy of breeding value estimates in South African Holstein cattle. We specifically tested modifications such as blending, tuning, and scaling to see how well genomic and pedigree information can be combined. Our findings revealed that ssGBLUP was more accurate than traditional pedigree-based methods, but this accuracy was influenced by how genomic and pedigree data were adjusted. Blending strategies with up to 40% polygenic effects increased prediction accuracy. Tuning methods had a less significant impact, and no tuning occasionally led to optimal performance. Scaling adjustments influenced prediction accuracy, with some lower scaling options resulting in improved accuracy for certain traits. This study shows the importance of parameters used in genomic prediction models to ensure more reliable genetic evaluations, which will ultimately assist farmers in making more accurate breeding decisions and increasing dairy production in South Africa.

## 1. Introduction

Predictions of genomic estimated breeding values (GEBVs) using single-nucleotide polymorphism (SNP) markers together with pedigree and phenotypic information provide the information needed for accurate selection of breeding animals [1,2,3,4,5]. Benefits include increasing the rate of genetic progress, reducing costs, and facilitating the use of animals from different groups without pedigree relationships [6,7,8,9]. Genomic prediction accuracy is dependent on factors such as the number of genotypes, relationships among genotyped animals, the density of SNP markers, statistical methods, linkage disequilibrium (LD), heritability, and genetic structure of the traits [10,11,12]. Most importantly, the genomic prediction model is a key factor that affects the prediction accuracy [12,13]. Single-step genomic best linear unbiased prediction (ssGBLUP) is commonly used in many practical applications and considered superior compared to the conventional pedigree-focused best linear unbiased prediction (ABLUP) methods [1,2,3,4,5,14,15].

In ssGBLUP, the inverse of the pedigree relationship matrix (**A**^−1^) is replaced by the inverted realized relationship matrix (**H**^−1^) [16]. The **H**^−1^ is calculated from **A**^−1^, the inverse of the genomic relationship matrix (**G**^−1^), and the inverse of the pedigree relationship matrix for genotyped animals ($A_{22}^{-1}$). However, ssGBLUP overlooks factors such as variability of the SNP effects and variation across genomic regions [17]. These issues can be addressed through the weighted ssGBLUP, which assigns different weights to SNP based on their estimated contributions to genetic variance and has shown potential in numerous studies despite potential effects on the dispersion of breeding values and bias [14,18,19]. Furthermore, GEBVs obtained from ssGBLUP are associated with bias linked to various factors, including singularity of the **G** matrix, incompatible **G** and **A**_22_ matrices, non-random selection of genotyped animals, and allele frequencies used in the **G** matrix creation [15,20,21,22,23,24]. Therefore, it is important to establish strategies to reduce the influence of these factors to improve the genomic prediction accuracy of ssGBLUP. Some of the strategies used to reduce bias and improve the accuracy of genomic predictions when merging the **G** and **A**_22_ include blending, tuning, and scaling [24,25,26].

Blending conditions of the **G** and **A**_22_ ensure that **G** is not singular and the relationship matrix is positive definite [15,23,27]. Tuning ensures that **A**_22_ is consistent by rebasing and scaling **G**, which corrects for differences in the variability in elements of **G** and **A**_22_ [15,22,28]. Lastly, scaling restricts **G** and **A**_22_ to minimize over- or under-estimation of the GEBVs [15,23]. Although the BLUPF90 software applies default blending and tuning parameters, these values may not be universally optimal. Their impact on evaluations can be significant, particularly in populations with limited genotyping or incomplete pedigree information, where inappropriate values may exacerbate inconsistencies between **G** and **A**_22_ or introduce additional bias [15,22,23,29]. Thus, exploring the most suitable adjustments related to the construction of the **H**^−1^ matrix of the ssGBLUP is essential for addressing bias and reducing the inflation/deflation of the GEBVs, thereby improving their accuracy.

In the absence of substantial information regarding genomic prediction in South African Holstein cattle, there is limited information as to how adjustments to the **H**^−1^ matrix could affect the accuracy of the GEBVs. Therefore, the objective of this study was to evaluate the accuracy of GEBVs predicted using ssGBLUP with standard blending, tuning, and scaling parameters, and compare these to alternative parameter configurations applied independently to the inverted relationship matrix for milk production traits in South African Holstein cattle.

South African Holsteins present a unique population for genomic evaluation, characterized by a historical reliance on pedigree-based evaluations [30]. Despite recent national initiatives such as the Dairy Genomic Programme (DGP), which aims to expand genotyping efforts [31,32], progress has been limited by financial and infrastructural constraints. As a result, the number of genotyped animals remains low compared to those in developed countries. Consequently, this study highlights the importance of testing ssGBLUP strategies to determine how parameter adjustments affect prediction accuracy in South African Holstein cattle with small genotypes.

## 2. Materials and Methods

### 2.1. Data Sources and Editing

Phenotypic and pedigree data were obtained from the national database, the Integrated Registration and Genetic Information System, South Africa. The original pedigree data included 3,699,231 data points for the Holstein cattle. The phenotype data consisted of 305-day lactation yield records for 4,779,369 cows. Dairy traits considered were milk, protein, and fat yields. Only the first three lactation records from 1989 to 2016 were used in the analysis; however, only cows with a first lactation record were retained. Cows without a recorded first lactation, even if they had records for later lactations, were excluded.

The pedigree data were edited to exclude records with unknown birth and calving dates. Age at calving was restricted from 20 to 42, 30 to 54, and 40 to 67 months for lactations 1, 2, and 3, respectively [33,34]. Records of milk yields below 1000 kg and more than 30,000 kg, as well as records of butterfat and protein percentages less than 2% or greater than 9%, were also excluded. The data were edited further to remove incomplete lactation records and those deemed unusable for genetic evaluations [33,34].

The edited phenotypic data used for the analysis contained 696,413 milk production records from 354,228 cows across 1991 herds. The final pedigree data comprised 541,325 animals: 9355 sires and 328,929 dams. Descriptive statistics of the phenotypic data are presented in Table 1. The season variable was categorized based on calving periods, with summer defined as October to March and winter as April to September. These two distinct seasons were used to account for potential environmental and management differences affecting lactation performance. Cows were assigned to contemporary groups defined by herd-year-season of calving. Contemporary groups with fewer than five animals and/or fewer than two sires were excluded, resulting in 22,410 groups.

### 2.2. Genotypic Data

Genomic data were generated through the DGP, which is a consortium between the ARC, South African universities, and the dairy industry, funded by the South African government. The Illumina 50K chip v3 (Illumina Inc., San Diego, CA, USA), featuring 53,218 SNP markers, was used for genotyping 1473 Holstein cattle. These animals are registered in the INTERGIS database and were sampled through routine evaluations conducted by the DGP. Using PLINK v. 1.07 [35], uninformative markers with MAF < 0.05 and markers with low genotyping rate were removed, as well as animals with an individual call rate < 0.90. Markers that deviated from the Hardy–Weinberg equilibrium (*p* < 0.0001) were also removed. This resulted in 1221 genotyped animals characterized by 41,407 SNP markers. The genotyped animals included 78 bulls and 1143 cows. Pedigree records indicated that 1218 animals had both parents recorded, while only 3 animals had a single recorded parent.

### 2.3. Statistical Analysis

Two approaches were used to predict breeding values, namely ABLUP and ssGBLUP. A pedigree-based relationship matrix was used in ABLUP, while the ssGBLUP model incorporated the **H** matrix. Computations were calculated using the BLUPF90 family of programs [36].

#### 2.3.1. Pedigree-Focused Best Linear Unbiased Prediction

The pedigree-based ABLUP model was utilized to estimate variance components and estimated breeding values (EBVs) for milk, protein, and butterfat yields. Variance components were estimated using the average information restricted maximum likelihood method, implemented through AIREMLF90 v1.149. The resulting heritability estimates are shown in Table 1.

Subsequently, EBVs were predicted using BLUPF90 v1.63, which implements the Best Linear Unbiased Prediction (BLUP) model in the BLUPF90 family of programs. The following single-trait repeatability model was used for the estimations:**y = Xb + Za + Wpe + e**(1)
where y is the vector of observations for the traits; **X**, **Z**, and **W** are the known incidence matrices relating records to fixed, random, and permanent environmental effects, respectively; **b** is the vector of fixed effects (herd-year-season, age at calving, parity); **a** is the vector of additive genetic effects for each animal, following a normal distribution **N**(0,**A**$\sigma_{a}^{2}$), with **A** as a pedigree-based additive genetic relationship matrix and $\sigma_{a}^{2}$ as the additive genetic variance; **pe** is the vector of permanent environmental effects following a normal distribution *N* (0, I$\sigma_{pe}^{2}$), and $\sigma_{pe}^{2}$ is the permanent environmental variance; and **e** is the vector of residual effects, following a normal distribution *N* (0, I$\sigma_{e}^{2}$), with $\sigma_{e}^{2}$ as the residual variance and **I** as the identity matrix.

#### 2.3.2. Single-Step Genomic Best Linear Unbiased Prediction

The GEBVs were estimated using the single-trait repeatability ssGBLUP model, like the ABLUP model. However, in ssGBLUP, the relationship matrix is replaced by the **H** matrix [2,16,23], which combines genotypes and pedigree data. Thus, the inverse of the matrix **H** is the following:(2)$$H^{-1}=A^{-1}+\left[\begin{array}{cc} 0 & 0\\ 0 & \tau G^{-1}-\omega A_{22}^{-1} \end{array}\right]$$where **A**^−1^ is the inverse of the numerator relationship matrix (**A**), including all animals; $G^{-1}$ is the inverse of the genomic relationship matrix; $A_{22}^{-1}$ is the inverse of the **A** matrix for only genotyped animals; and weighting factors for $G^{-1}$ and $A_{22}^{-1}$ are represented by the ***τ*** and ***ω***, respectively. **G** is created according to VanRaden [20]. Adjustments of the **H** matrix inverse were explored to assess the influence of blending, tuning, and scaling factors on prediction accuracies.

##### Blending

In ssGBLUP, the blending of the **G** and **A**_22_ matrices was carried out as (1 − β)$G^{-1}$ + β$A_{22}^{-1}$, where β represented the amount of residual polygenic variance unaccounted for by **G**. In the current study, the blending strategies were defined by varying β = 0.05, 0.10, 0.20, 0.30, and 0.40 and abbreviated as ssGBLUP_G0.95, ssGBLUP_G0.90, ssGBLUP_G0.80, ssGBLUP_G0.70, and ssGBLUP_G0.60, respectively. In addition, β = 0.05 was benchmarked as a standard blending coefficient that has been commonly used in ssGBLUP applications. However, β values as low as 0.50 may be appropriate when **G** captures most of the additive genetic variance [23,37].

##### Tuning

Options available for tuning in the BLUPF90 program were explored to determine their impact on predictions. Tuning was accomplished by adjusting the values of **G** and **A**_22_ as follows: ssGBLUP_TG0 = no adjustment; ssGBLUP_TG1 = Mean(diag(**G**)) = 1 and Mean(offdiag(**G**)) = 0 proposed by Legarra [38], where diag is the diagonal elements and offdiag is the off-diagonal elements; ssGBLUP_TG2 = Mean(diag(**G**)) = Mean(diag(**A**_22_)) and Mean(offdiag(**G**)) = Mean(offdiag(**A**_22_)) proposed by Chen et al. [28] and referred to as standard; ssGBLUP_TG3 = Mean(**G**) = Mean(**A**_22_) [22]; and ssGBLUP_TG4 = rescaling **G** using an Fst adjustment [22,39].

##### Scaling

Scaling was implemented in ssGBLUP with the scaling factors ***τ*** and ***ω*** as shown in Equation (2). The accuracy of prediction using ssGBLUP was evaluated by varying the scaling parameters ***τ*** and ***ω***. The scaling values tested for both parameters were 0.60, 0.70, 0.80, 0.90, and 1.0, with the standard BLUPF90 default set at 1.0. Each combination of ***τ*** and ***ω*** was applied separately to assess their impact on genomic prediction accuracy.

The ssGBLUP model was implemented independently for each scaling value, resulting in 10 analyses: five analyses with varying ***τ*** while keeping ***ω*** = 1.0, and five analyses with varying ***ω*** while keeping ***τ*** = 1.0.

#### 2.3.3. Validation of Prediction Accuracy

To assess the accuracy, a forward validation approach was implemented. Two datasets were used for evaluations: (1) the complete set of phenotype records and (2) datasets with phenotypic records for 390 genotyped cows intentionally omitted. These 390 cows were selected from a total of 1221 genotyped animals. Cows whose phenotype records were omitted had at least one lactation record, and priority was given to those from more recent births. The 831 remaining genotyped animals, in combination with the phenotypic and pedigree data from both genotyped and ungenotyped animals, formed the basis for predicting the breeding values of the 390 validation cows.

This approach differs from cross-validation, which divides genotyped animals into several subsets for iterative training and validation. Due to the relatively small number of genotyped animals available in our study, forward validation was chosen instead. Forward validation closely mirrors the practical implementation of genetic evaluations in livestock populations [40].

The terminology “realized accuracy” was used to indicate the Pearson correlation of the solutions for the 390 cows from the paired mixed model analyses that either do or do not contain their phenotypic data. The formula used to compute realized accuracy is as follows:Realized Accuracy = cor(EBV_BLUP_full_, G/EBV_Reduced_)(3)
where EBV_BLUP_full_ refers to the EBVs estimated using the full BLUP model with all phenotypes, and G/EBV_Reduced_ refers to the EBVs or GEBVs estimated without phenotypic records for the 390 validation cows.

## 3. Results

### 3.1. Genomic Prediction Accuracy of ABLUP and ssGBLUP Models

Presented in Table 2 are the prediction accuracies from ABLUP and ssGBLUP models for milk, protein, and fat. Use of ssGBLUP produced large increases in accuracy relative to ABLUP. The mean individual accuracies across all models ranged from 0.58 to 0.61. This result shows the advantage of incorporating genomic information in genetic evaluations.

### 3.2. Accuracy of Predictions Using Different Blending Parameters

Table 3 shows that accuracy increased as the blending parameter (β) increased. The accuracy for milk, protein, and fat decreases slightly as the genomic parameter increases, with values for milk ranging from 0.26 at ssGBLUP_G0.60 to 0.23 at ssGBLUP_G0.95, protein ranging from 0.32 to 0.29, and fat ranging from 0.33 to 0.30. These results suggest that while genomic information improves accuracy, excessive reliance on genomic relationships may diminish predictive power.

### 3.3. Accuracy of Predictions Using Different Tuning Options

The prediction accuracy of ssGBLUP was improved at the lowest value of the tuning parameter (Table 4). The accuracy for milk, protein, and fat is highest at ssGBLUP_TG0 and ssGBLUP_TG1, with values of 0.25 for milk, 0.30 for protein, and 0.31 for fat, and decreases slightly from ssGBLUP_TG2 (default) onwards, where the values stabilize at 0.23 for milk, 0.29 for protein, and 0.30 for fat. This indicates that minimal tuning of the genomic relationship matrix can enhance prediction accuracy, whereas aggressive adjustments may reduce the reliability of genomic estimates.

### 3.4. Accuracy of Predictions Using Different Scaling Parameters for **τ** and **ω**

The results for the ***τ*** and ***ω*** scaling factors are presented in Table 5. The accuracy for milk ranges from 0.22 at ***τ*** 0.60 and ***τ*** 0.70 to 0.23 from ***τ*** 0.80 onwards. The accuracy increases from 0.28 at ***τ*** 0.60 to 0.29 at ***τ*** 0.70, while remaining stable through ***τ*** 1 (default) for protein. For fat, the accuracy increases from 0.28 at ***τ*** 0.60 to 0.30 at ***τ*** 0.90 and remains at 0.30 at ***τ*** 1. These findings suggest that higher ***τ*** values help maintain predictive stability, likely by balancing pedigree and genomic contributions effectively.

In terms of scaling ***ω***, the accuracy for milk decreases from 0.26 at ***ω*** 0.60 and ***ω*** 0.70 to 0.23 at ***ω*** 1 (default). For protein, the accuracy declines from 0.32 at ***ω*** 0.60 and ***ω*** 0.70 to 0.29 at ***ω*** 1. For fat, the accuracy decreases from 0.34 at ***ω*** 0.60 to 0.30 at ***ω*** 1. The decline in accuracy at higher ***ω*** values suggests that while genomic information is beneficial, overemphasizing it relative to pedigree data may compromise predictive precision.

## 4. Discussion

The current study found that ssGBLUP models outperformed ABLUP in predicting EBVs in Holstein cattle. Generally, traditional BLUP models achieved low realized accuracies due to the limited and perhaps inaccurate parentage in pedigree records, which could explain the low prediction accuracies [41]. Similar low realized accuracies have been reported, with ABLUP as low as 0.12 and ssGBLUP at 0.23 [42], and ABLUP accuracy as low as 0.02 [43]. These studies confirm that low ABLUP or ssGBLUP accuracies are not unique to the current study and can occur when data limitations exist.

The correlations between the EBVs and the GEBVs for the full model are presented in Appendix A. As expected, there were strong correlations (0.86–0.88) between EBVs and GEBVs. However, the mean individual accuracies across models were identical (e.g., 0.61 for milk). This shows that pedigree-based ABLUP may exaggerate EBV confidence [44], particularly for cows, which often have lower accuracies due to fewer or no progeny [45,46,47,48]. The low ABLUP accuracies (Table 2) were most likely caused by the under-representation of bulls, typically more informative in genetic evaluations, and the predominance of younger over older animals. As a result, inadequate pedigree connectedness, limited progeny data, and previous selection all lower parent average accuracy [49]. ABLUP, which lacks genomic data, cannot overcome these constraints when compared to ssGBLUP, which uses both pedigree and genomic information to increase predictive accuracy [50].

This study reiterates the importance of using genomic data through ssGBLUP, which integrates genotyped and non-genotyped animals in a single evaluation [5,11,16,51,52]. However, the realized accuracies found in the current study were generally low but consistent with the ranges reported in previous research (9% to 47%) using small numbers of genotypes in ssGBLUP for traits such as milk yield in the Holstein breed [45,46,47,48]. It is indeed remarkable that having genotypes for only a relatively small fraction of the animals produced improvements in prediction accuracy of this magnitude.

Despite ssGBLUP consistently outperforming ABLUP in this study, the low realized accuracy observed is likely due to the limited number of genotyped animals, the degree of relatedness among them, and the characteristics of the prediction models used [10,11,12]. While ssGBLUP was utilized in this study, the GEBVs obtained are associated with potential biases resulting from the singularity of **G** and challenges associated with compatibility between **G** and **A**_22_, which could lead to low genomic prediction accuracy [5,20,22,24].

The present study utilized genotypes from medium-density SNP chip panels (50K), which, while not capturing the entire genome, can effectively cover LD blocks for genomic relationship matrix construction [53]. Therefore, studying alternative strategies for blending pedigree and genomic information is necessary to evaluate different amounts of information being captured by polygenic effects [53]. This study shows that prediction accuracy improves when polygenic effects account for up to 40% of the information used in calculating the GEBVs for Holstein cattle, as seen with the ssGBLUP models using blending values of β = 0.40, β = 0.30, and β = 0.20. This contrasts with the standard value of β = 0.05 used in the ssGBLUP_G0.95 model. Our results align with those of Curzon et al. [54], where increasing β was beneficial for a small Israeli Holstein population, and giving more weight to pedigree (β = 0.30–0.50) reduced the upward bias of GEBVs and improved prediction accuracy. According to Lourenco et al. [23], increasing β is often necessary to control inflation when pedigree data are incomplete and can speed up convergence with minimal or no impact on accuracy. In our study, pedigree completeness was generally high among the genotyped animals, with 1218 out of 1221 animals having both parents recorded, indicating that adjustments in β were likely not driven by pedigree gaps but by other structural aspects of the data and model.

Although β = 0.05 is commonly used for blending to address singularity issues in BLUPF90 programs [23], the results of this study demonstrate variations in prediction accuracy when using alternative blending values. Consistent with Piccoli et al. [53], the current study shows that β = 0.05 provides reliable accuracy, and only a marginal improvement in accuracy resulted from using other values for blending. Interestingly, while Gao et al. [55] and Neves et al. [56] reported gains with β = 0.20, the present study shows limited advantage when using this value, suggesting that the optimal blending parameter may be population-dependent. Furthermore, in agreement with Abdalla et al. [27], the results of this study reveal minimal differences across blending strategies, although β = 0.05 and β = 0.10 emerge as practical choices. These findings highlight the importance of polygenic effects, which, as Meyer et al. [57] noted, can provide up to 20% of the relationship information among animals. In this population study, weighting the polygenic effects more heavily improved accuracy when marker-based relationships alone were insufficient to capture additive genetic variance. This means that increasing the reliance on polygenic effects depends on certain conditions and populations [57,58,59]. Overall, these results indicate that the optimal blending value is context-dependent, influenced by population structure, relatedness, and the proportion of information contributed by pedigree and genomic data. Although the current study did not evaluate pedigree depth or accuracy beyond parentage, these factors may influence the optimal blending value in other populations [23,54]. Therefore, developing general strategies for determining appropriate β values under different pedigree structures warrants further investigation, particularly in larger datasets with variable pedigree depth.

Moreover, the effectiveness of blending, and ssGBLUP more broadly, also hinges on the composition of the genotyped reference population. The limited number of genotyped animals (1221) relative to the large population with phenotypes restricts the informativeness of the genomic relationship matrix and thus ssGBLUP accuracy [16,60]. Selecting animals to include in the genotyped reference population is crucial to maximize connectedness, genetic diversity, and family representation so as to improve prediction accuracy [61,62]. Beyond the work of Spangler et al. [63], future efforts should improve on genotyping strategies to enhance genomic evaluations, especially in settings with limited resources.

Exploring additional ways to modify **H**^−1^ may result in improved accuracy of prediction. Previously, tuning and scaling have been identified as being crucial [59]. Tuning ensures compatibility between **G** and **A**_22_ matrices, and both refer to the same genetic base [37]. Nevertheless, the explored tuning methods made only small changes to the realized accuracy compared to setting the means of the off-diagonal and diagonal elements of **G** to the means of the off-diagonal and diagonal elements of **A**_22_, which is standard in BLUPF90 programs [28,64]. McWhorter et al. [37] also found minor differences in accuracy from various tuning options. According to Neshat et al. [65], tuning improves the genomic prediction accuracy based on their research using simulated genomic data. Consequently, some tuning methods that depend on the allele frequencies, such as equating the means of the off-diagonal and diagonal elements of **G** and **A**_22_ or rescaling **G** by F_st_, are computationally demanding [65]. Hence, a tuning strategy that does not depend on allele frequencies with simplified computations was proposed by Bermann et al. [64]. However, it did not perform better than the above-mentioned standard strategies. In the present study, the option of not tuning seemed better than the standard method, although not to a large extent. There is limited evidence to justify this observation based on studies that considered not tuning as an option. Bermann et al. [64] and Hsu et al. [66] demonstrated lower accuracy of GEBVs without tuning compared to tuning. Consequently, it is important to recognize that not tuning may inflate and bias the GEBVs [64]. Further investigation into the no-tuning approach is warranted, particularly when combined with adjustments to the **H**^−1^ matrix, such as blending and scaling. This will help identify the most effective combination of tuning methods, which will be explored in a subsequent article.

One of the challenges in ssGBLUP is that when the G^−1^ and $A_{22}^{-1}$ matrices are not properly scaled in the **H**^−1^ matrix, inflated or deflated GEBVs may result [15]. To address this, ***ω*** scaling regulates $A_{22}^{-1}$ while ***τ*** regulates **G**^−1^ [23]. This makes it necessary to ensure proper scaling of genomic and pedigree matrices using ***τ*** and ***ω*** values [23]. The findings of this study indicate that the scaling values of ***τ*** and ***ω*** have a significant effect on the realized accuracy of EBVs estimated with ssGBLUP. Adjusting ***τ*** to a value less than 1 did not improve the realized accuracy in Holstein, contrary to previous research [5,67]. This study focused on values of ***τ*** less than 1, following the basis established in prior research. However, previous studies have demonstrated that increasing ***τ*** values beyond 1 can significantly improve prediction accuracy [68,69]. The decision to limit the exploration to values less than 1 was a limitation of this study. Further exploration of higher ***τ*** values could reveal additional benefits.

This study found that reducing ***ω*** to 0.6 improved the realized accuracy in the evaluation. Similar trends were observed in other studies when ***ω*** was reduced to less than 1 [5,67,68,69]. Conversely, Hong et al. [70] observed reduced prediction accuracy with ***ω*** less than 1. According to Lourenco et al. [23], reducing ***ω*** below 1 contributes to reducing overestimation of the GEBVs. The current results emphasize the need for careful consideration of population-specific factors and thorough evaluation of scaling parameters in genomic prediction analyses.

Scaling ***ω*** down to 0.60 and blending with up to 20% polygenic effects enhanced Holstein prediction accuracy, emphasizing the significance of these modifications in the matrices that were derived from the relationships among animals. Aligning the **G**^−1^ with $A_{22}^{-1}$ has improved accuracy and reduced bias in GEBVs [71]. Therefore, these studies indicate a more effective accounting for additive genetic variation. However, the necessity of fine-tuning was recommended to account for higher proportions of blending or when the model explicitly accounts for a residual polygenic effect [29,72]. Further research needs to focus on optimizing the tuning and blending of additive effects, particularly when exceeding 20%, and explore the benefits of incorporating ancestral genetic contributions through pedigree information to enhance prediction accuracy and reduce bias in genomic prediction models.

Given that the accuracy achieved with the current approach remains low, additional strategies are needed to enhance it. While the focus here was on ssGBLUP, the next step would involve exploring how combinations of parameters such as blending, scaling, and no-tuning could improve prediction accuracy and reduce bias, an aspect not fully addressed in this study. It is also significant to determine how these affect bias due to the inflation of differences among the GEBVs. Neshat et al. [59] recommended the importance of establishing the best configuration of blending, tuning, and scaling parameters to improve prediction accuracy. However, there are arguments around the best way to combine these parameters when adjusting the **H**^−1^ matrix [37,70,73]. These adjustments are crucial, as they optimize the blending of pedigree and genomic information and directly impact the accuracy of GEBVs, leading to more informed breeding decisions and achieving genetic progress in livestock populations.

## 5. Conclusions

Single-step GBLUP outperformed ABLUP across all traits. However, overall prediction accuracies remained low, likely influenced by factors such as the limited number of genotyped animals, the structure of the reference population, and model assumptions. Blending with up to 40% polygenic effects slightly improved accuracy, which means that the optimal blending parameter is population-dependent. Tuning methods showed only marginal improvements, and the no-tuning approach produced slightly better results, suggesting the need for further exploration of no-tuning strategies combined with blending and scaling. Scaling parameters ***ω*** had a significant effect on accuracy, especially when reduced. These findings highlight the influence of blending, tuning, and scaling choices to enhance genomic prediction accuracy. Therefore, it is recommended that preliminary optimization be tailored to a specific population and that data be collected before genomic evaluations. As this study is preliminary and constrained by a limited number of genotyped animals, it is recommended that the SA Holstein Cattle Breeders’ Society and breeders prioritize genotyping not only more animals but also those that are more informative to enhance the accuracy and reliability of future genomic evaluations.

## Figures and Tables

**Table 1 animals-15-02866-t001:** Characteristics of the data used to assess the accuracy of genomic prediction with different parameters for blending, tuning, and scaling.

Trait	Descriptive Statistics			Heritability
	Minimum	Maximum	Mean ± SD	
Milk yield (kg)	1000	25,993	7940.10 ± 2615.10	0.28
Protein (kg)	25	857.19	290.23 ± 100.25	0.21
Fat (kg)	26	833.88	252.54 ± 82.98	0.25

Standard deviation (SD).

**Table 2 animals-15-02866-t002:** Comparison of the realized accuracy of the EBV from pedigree-based BLUP (ABLUP) and the single-step GBLUP (ssGBLUP) analyses.

Model	Milk	Protein	Fat
ABLUP	0.01	0.03	0.03
ssGBLUP	0.23	0.29	0.30

ABLUP = Traditional animal model best linear unbiased prediction; ssGBLUP = Single-step genomic best linear unbiased prediction.

**Table 3 animals-15-02866-t003:** Effect of different genetic relationship matrix blending approaches on prediction accuracy.

Model	Milk	Protein	Fat
ssGBLUP_G0.60	0.26	0.32	0.33
ssGBLUP_G0.70	0.26	0.32	0.33
ssGBLUP_G0.80	0.25	0.31	0.32
ssGBLUP_G0.90	0.24	0.30	0.30
ssGBLUP_G0.95	0.23	0.29	0.30

ssGBLUP = Single-step genomic best linear unbiased prediction; G = Genomic relationship matrix blending proportion. The models ssGBLUP_G0.60 to ssGBLUP_G0.95 represent blending strategies with β values of 0.40, 0.30, 0.20, 0.10, and 0.05, respectively.

**Table 4 animals-15-02866-t004:** The realized accuracy of the EBV from single-step GBLUP analyses using different tuning options in the construction of the inverse of the combined pedigree and genomic relationship matrix.

Model	Milk	Protein	Fat
ssGBLUP_TG0	0.25	0.30	0.31
ssGBLUP_TG1	0.25	0.30	0.31
ssGBLUP_TG2	0.23	0.29	0.30
ssGBLUP_TG3	0.23	0.29	0.30
ssGBLUP_TG4	0.23	0.29	0.30

ssGBLUP_TG0 = No scaling; ssGBLUP_TG1 = mean(diag(**G**)) = 1 and mean(offdiag(**G**)) = 0; ssGBLUP_TG2 = mean(diag(**G**)) = mean(diag(**A**_22_)) and mean(offdiag(**G**)) = mean(offdiag(**A**_22_)); ssGBLUP_TG3 = mean(**G**) = mean(**A**_22_); ssGBLUP_TG4 = rescale **G** using an Fst adjustment.

**Table 5 animals-15-02866-t005:** Effect of different parameter scaling strategies (***τ*** and ***ω***) on prediction accuracy.

	Model	Milk	Protein	Fat
Scaling ***τ***	ssGBLUP_***τ*** 0.60	0.22	0.28	0.28
	ssGBLUP_***τ*** 0.70	0.22	0.29	0.29
	ssGBLUP_***τ*** 0.80	0.23	0.29	0.29
	ssGBLUP_***τ*** 0.90	0.23	0.29	0.30
	ssGBLUP_***τ*** 1	0.23	0.29	0.30
Scaling ***ω***	ssGBLUP_***ω*** 0.60	0.26	0.32	0.34
	ssGBLUP_***ω*** 0.70	0.26	0.32	0.33
	ssGBLUP_***ω*** 0.80	0.25	0.31	0.32
	ssGBLUP_***ω*** 0.90	0.25	0.30	0.31
	ssGBLUP_***ω*** 1	0.23	0.29	0.30

Scaling ***τ*** = different parameter scaling strategies for ***τ***; Scaling ***ω*** = different parameter scaling strategies for ***ω***.

## Data Availability

The data analyzed in this study were obtained from the Dairy Genomic Programme and the SA Holstein Cattle Breeders’ Society. These datasets are not publicly available because they are owned by third parties. Data may be made available for research purposes upon reasonable request to the corresponding author.

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
