# Peer review of "Preliminary Evaluation of Blending, Tuning, and Scaling Parameters in ssGBLUP for Genomic Prediction Accuracy in South African Holstein Cattle"

_animals, 2025, doi:10.3390/ani15192866_

Round 1

Reviewer 1 Report (New Reviewer)

Comments and Suggestions for Authors

The data and methods are clearly presented.

The main outcome is that blending increases the accuracies of EBVs.

The proportion of pedigree data deems to have an optimum.

The authors should work out more general rules how to determine this optimum.

This optimum may depend on pedigree depth and accuracies of pedigrees.

This issue has to be worked in much more detail.

Another point is the size of the data set.

Does this influence the outcomes.

Have you a large data set for the breeding population and could you compare your genotyped data and ssGBLUP EBVs for such a large dataset.

The issue how to select animals for genotyping is a further important issue.

This issue has also to be worked out and has also to be discussed.

Author Response

Response to Reviewer 1 Comments

1. Summary

We would like to express our sincere gratitude for your careful review and constructive comments on our manuscript. Your thoughtful suggestions and professional insights have been extremely valuable in improving the clarity, depth, and overall quality of our work. In response to your feedback, we have thoroughly revised the manuscript. Please find our detailed responses to each point below, with all corresponding changes clearly highlighted in the revised manuscript text. We have addressed all comments carefully and provided additional explanation or justification where appropriate. Thank you once again for your time, effort, and invaluable input.

2. Questions for General Evaluation

Reviewer’s Evaluation

Response and Revisions

Does the introduction provide sufficient background and include all relevant references?

Yes

Is the research design appropriate?

Can be improved

Are the methods adequately described?

Can be improved

Are the results clearly presented?

Yes

Are the conclusions supported by the results?

Can be improved

Are all figures and tables clear and well-presented?

Yes

3. Point-by-point response to Comments and Suggestions for Authors

Comments 1: The data and methods are clearly presented.

Response 1: Thank you for the positive feedback. We appreciate the reviewer’s comment noting the clarity of the data and methods presented in the manuscript.

Comments 2: The main outcome is that blending increases the accuracies of EBVs.

Response 2: Thank you for highlighting the main outcome. We confirm that blending polygenic effects up to 40% improved prediction accuracy in this study, which aligns with findings in other small or moderately sized reference populations. This supports the importance of optimizing blending parameters for improved genomic prediction under limited genotyping scenarios.

Comments 3: The proportion of pedigree data deems to have an optimum.

The authors should work out more general rules how to determine this optimum.

This optimum may depend on pedigree depth and accuracies of pedigrees.

This issue has to be worked in much more detail.

Response 3: Thank you for this important suggestion. We have clarified this issue in the revised manuscript (L367–371): “Although the current study did not evaluate pedigree depth or accuracy beyond parentage, these factors may influence the optimal blending value in other populations [23,54]. Therefore, developing general strategies for determining appropriate β values under different pedigree structures warrants further investigation, particularly in larger datasets with variable pedigree depth.”

Comments 4: Another point is the size of the data set.

Does this influence the outcomes.

Have you a large data set for the breeding population and could you compare your genotyped data and ssGBLUP EBVs for such a large dataset.

The issue how to select animals for genotyping is a further important issue.

This issue has also to be worked out and has also to be discussed.

Response 4: Thank you for highlighting this important issue. We have clarified and expanded on this in the revised manuscript (L372–380):

“Moreover, the effectiveness of blending and ssGBLUP more broadly also hinges on the composition of the genotyped reference population. The limited number of geno-typed animals (1,221) relative to the large population with phenotype restricts the in-formativeness of the genomic relationship matrix and thus ssGBLUP accuracy [16,60]. Selecting animals to include in the genotyped reference population is crucial to max-imize connectedness, genetic diversity, and family representation so as to improve prediction accuracy [61,62]. Beyond the work of Spangler et al. [63], future efforts should improve on genotyping strategies to enhance genomic evaluations, especially in settings with limited resources. ”

4. Response to Comments on the Quality of English Language

Point 2: The English is fine and does not require any improvement.

Response 1: Thank you for the positive feedback. We have reviewed the manuscript again and made minor language edits where necessary to improve clarity and consistency, while retaining the overall tone and flow.

5. Additional clarifications

Reviewer 2 Report (Previous Reviewer 2)

Comments and Suggestions for Authors

Please response all the comments below:

L77. Since the values are not universal, do you suggest all researcher to optimize parameters before performing the real analysis? Please state clearly in the conclusion.
L163-164. Missing variance term. The author missed N(0, I*var(pe)) and Var(pe) as the permanent environmental variance.
L249. The realized accuracy is quite high. I would like to see the correlation between ACC of BLUP_BV and ACC of GEBV?
L263. The result in table 3 seems conflict with abstract about whether 40% blending yields the highest accuracy.
273. The result of no tuning gave the best result seems contradict to previous reports. Please justify this more clearly.
L302. Table 5. Why the author choose only τ less than 1, as the incresasin values showed to improve accuracy.

Thank you.

Author Response

Response to Reviewer 2 Comments

1. Summary

We would like to express our sincere gratitude for your careful review and constructive comments on our manuscript. Your thoughtful suggestions and professional insights have been extremely valuable in improving the clarity, depth, and overall quality of our work. In response to your feedback, we have thoroughly revised the manuscript. Please find our detailed responses to each point below, with all corresponding changes clearly highlighted in the revised manuscript text. We have addressed all comments carefully and provided additional explanation or justification where appropriate. Thank you once again for your time, effort, and invaluable input.

2. Questions for General Evaluation

Reviewer’s Evaluation

Response and Revisions

Does the introduction provide sufficient background and include all relevant references?

Yes

Is the research design appropriate?

Can be improved           

Are the methods adequately described?

Can be improved

Are the results clearly presented?

Can be improved

Are the conclusions supported by the results?

Yes

Are all figures and tables clear and well-presented?

Yes

3. Point-by-point response to Comments and Suggestions for Authors

Comments 1: L77. Since the values are not universal, do you suggest all researcher to optimize parameters before performing the real analysis? Please state clearly in the conclusion.

Response 1: Thank you for your helpful suggestion. We changed the conclusion to make it clear that researchers should optimise blending, tuning, and scaling parameters based on their specific population and dataset features before analysing them, as ideal values are population-dependent and crucial for enhancing genetic prediction accuracy.

Comments 2: L163-164. Missing variance term. The author missed N(0, I*var(pe)) and Var(pe) as the permanent environmental variance.

Response 2: Thank you for pointing this out. The paragraph has been updated to include the variance structure for permanent environmental effects. The phrase is now translated as follows: "pe is the vector of permanent environmental effects following a normal distribution N (0, I,?-??-2.) and ,?-??-2. as the permanent environmental variance" This correction has been incorporated into the manuscript to ensure clarity and completeness.

Comments 3: L249. The realized accuracy is quite high. I would like to see the correlation between ACC of BLUP_BV and ACC of GEBV?

Response 3: Thank you for your comment. We have added a discussion supported by Supplementary Table S1 (L308–319), which presents the correlation between the individual accuracies of EBVs from the pedigree-based BLUP and GEBVs from ssGBLUP.

Comments 4: L263. The result in table 3 seems conflict with abstract about whether 40% blending yields the highest accuracy.

Response 4: Thank you for your insightful comment on the apparent disparity between the abstract and Table 3 results.

We validate that blending proportions with pedigree contributions of 30% (β = 0.30) and 40% (β = 0.40) produce fairly similar prediction accuracies, as shown in Table 3. The models ssGBLUP_G0.60 (corresponding to β = 0.40) and ssGBLUP_G0.70 (corresponding to β = 0.30) achieved the best accuracies with minimal differences.

To increase consistency, we changed the abstract and accompanying text to show that blending proportions up to 40% (β = 0.30 to 0.40) produce the highest accuracies. This language more closely represents the results in Table 3 and avoids indicating a single optimal blending value.

We believe that this clarification addresses your concern and improves the clarity of our findings.

Comments 5: 273. The result of no tuning gave the best result seems contradict to previous reports. Please justify this more clearly.

Response 5: Thank you for making such an important observation about the performance of tuning options. Previous research has consistently shown that appropriate tuning improves prediction accuracy. However, as described in the text (Lines 393 onwards), our findings revealed that no tuning (ssGBLUP_TG0) and basic centring (ssGBLUP_TG1) produced slightly higher accuracies than alternative tuning strategies. We addressed this during the conversation, citing various pertinent studies.

We acknowledge the limitations of our finding, particularly the possibility of inflation or bias when omitting tuning, and will pursue further investigation into combinations of tuning with other H⁻¹ adjustments in future work.

We hope this addresses your concern, and we are happy to enhance this section as needed.

Comments 6: L302. Table 5. Why the author choose only τ less than 1, as the incresasin values showed to improve accuracy.

Response 6: Thank you for observing the τ value range in Table 5.

As stated in Lines 408-414 of the manuscript, limiting the scaling parameter τ to values smaller than 1 was a study limitation. Previous research indicates that using τ < 1 can effectively align G with A22, particularly in datasets with small genotyped populations.

We have clearly indicated this as a limitation in the discussion and suggested that future research should consider the effect of larger τ values to explore the full potential of scaling adjustments in ssGBLUP.

We hope this clarification addresses your concern.  

4. Response to Comments on the Quality of English Language

Point 2: The English is fine and does not require any improvement.

Response 1: Thank you for the positive feedback. We have reviewed the manuscript again and made minor language edits where necessary to improve clarity and consistency, while retaining the overall tone and flow.

5. Additional clarifications

Reviewer 3 Report (Previous Reviewer 3)

Comments and Suggestions for Authors

In my view, similar to the issue mentioned previously, the dataset used in this study on genomic selection breeding in Holstein cattle is relatively limited. Expanding the data size is necessary to enhance the representativeness of the results.

Author Response

Response to Reviewer 3 Comments

1. Summary

We would like to express our sincere gratitude for your careful review and constructive comments on our manuscript. Your thoughtful suggestions and professional insights have been extremely valuable in improving the clarity, depth, and overall quality of our work. In response to your feedback, we have thoroughly revised the manuscript. Please find our detailed responses to each point below, with all corresponding changes clearly highlighted in the revised manuscript text. We have addressed all comments carefully and provided additional explanation or justification where appropriate. Thank you once again for your time, effort, and invaluable input.

2. Questions for General Evaluation

Reviewer’s Evaluation

Response and Revisions

Does the introduction provide sufficient background and include all relevant references?

Can be improved

Is the research design appropriate?

Can be improved

Are the methods adequately described?

Yes

Are the results clearly presented?

Can be improved

Are the conclusions supported by the results?

Can be improved

Are all figures and tables clear and well-presented?

Can be improved

3. Point-by-point response to Comments and Suggestions for Authors

Comments 1: In my view, similar to the issue mentioned previously, the dataset used in this study on genomic selection breeding in Holstein cattle is relatively limited. Expanding the data size is necessary to enhance the representativeness of the results.

Response 1: Thank you for the insightful observation. We agree that expanding the dataset—particularly the number of genotyped animals—would improve the robustness and representativeness of genomic predictions. As mentioned in the discussion section, this limitation reflects the current state of genetic resources in the South African dairy industry, where genotyping is currently limited due to cost and infrastructure constraints. Nonetheless, the study gives valuable insights in real-world, resource-constrained settings, as well as a benchmark for continued progress. As genotyping becomes more accessible, future research will benefit from the inclusion of larger and more diverse reference populations.

4. Response to Comments on the Quality of English Language

Point 2: The English is fine and does not require any improvement.

Response 1: Thank you for the positive feedback. We have reviewed the manuscript again and made minor language edits where necessary to improve clarity and consistency, while retaining the overall tone and flow.

5. Additional clarifications

Reviewer 4 Report (New Reviewer)

Comments and Suggestions for Authors

Evaluating the Impact of Blending, Tuning, and Scaling Factors in ssGBLUP on Genomic Prediction Accuracy in South African Holstein Cattle is an interesting article that challenges the stereotypical criteria for genomic evaluations, with the aim of achieving better accuracy by testing different parameters of genomic evaluation.

The weakness of the research is that the traditional BLUP (ABLUP) method has very low accuracy, and no specific and direct reason for this is mentioned in the article. Minor comments:

Table 1: h2 column must replace to another table, because h2 is not a phenotypic data.

Table1: please mention unit for each trait. Unit for milk is litre or Kg?

Table1: Mention please ow h2 is calculated? By what method? Number of Data? Your model was uni-variate or multi variate?

L190; Verify please 0.50 or 0.05?

Tbale2; why accuracy of ABLUP is very low for all traits? In animal breeding, blup animal model for milk traits has suitable accuracy in comparison with traditional models such as Sire model. Please provide possible reasons for very low accuracy.

L428-429: please provide some justification for this. How you arrived to this conclusion that relatedness or model characteristics influence accuracy of ABLUP? Provide some documents or calculated parameters of your analysis.

L438: Why is the emphasis on the limited number of genotyped animals? Is the effect of the number calculated or simulated? Or is it guessed? Or is it stated based on the bibliography of the subject?

Author Response

Response to Reviewer 4 Comments

1. Summary

We would like to express our sincere gratitude for your careful review and constructive comments on our manuscript. Your thoughtful suggestions and professional insights have been extremely valuable in improving the clarity, depth, and overall quality of our work. In response to your feedback, we have thoroughly revised the manuscript. Please find our detailed responses to each point below, with all corresponding changes clearly highlighted in the revised manuscript text. We have addressed all comments carefully and provided additional explanation or justification where appropriate. Thank you once again for your time, effort, and invaluable input.

2. Questions for General Evaluation

Reviewer’s Evaluation

Response and Revisions

Does the introduction provide sufficient background and include all relevant references?

Yes

Is the research design appropriate?

Yes

Are the methods adequately described?

Can be improved

Are the results clearly presented?

Can be improved

Are the conclusions supported by the results?

Can be improved

Are all figures and tables clear and well-presented?

Can be improved

3. Point-by-point response to Comments and Suggestions for Authors

Comments 1: Evaluating the Impact of Blending, Tuning, and Scaling Factors in ssGBLUP on Genomic Prediction Accuracy in South African Holstein Cattle is an interesting article that challenges the stereotypical criteria for genomic evaluations, with the aim of achieving better accuracy by testing different parameters of genomic evaluation.

Response 1: Thank you for the positive feedback. Our objective was indeed to explore alternative configurations of ssGBLUP through blending, tuning, and scaling to address the challenges of genomic evaluation in resource-constrained settings. We are pleased that the reviewer finds this approach valuable in advancing genomic prediction strategies for South African Holstein cattle.

Comments 2: The weakness of the research is that the traditional BLUP (ABLUP) method has very jnlow accuracy, and no specific and direct reason for this is mentioned in the article. Minor comments:

Response 2: We appreciate the reviewer’s observation. However, we would like to clarify that the discussion section already addresses the low accuracy observed with the ABLUP model. Specifically, we attributed the reduced accuracy to limited and potentially inaccurate pedigree information, which is a known limitation of traditional BLUP approaches. This point was also supported by references to similar findings in the literature. Nonetheless, to ensure clarity, we have made a slight adjustment in the discussion to emphasize this point more explicitly.

Comments 3: Table 1: h2 column must replace to another table, because h2 is not a phenotypic data.

Response 3: Thank you for this helpful observation. We acknowledge that heritability (h²) is not a phenotypic parameter. However, to maintain transparency without introducing redundancy, we have revised the table title to better reflect its broader content. Following guidance from a co-author, we have updated the title to:

“Table 1. Characteristics of the data used to assess accuracy of genomic prediction with different parameters for blending, tuning and scaling.”

This revised title allows the inclusion of both descriptive statistics and heritability estimates, while avoiding confusion about the nature of the data presented.

We hope this clarification resolves the concern.

Comments 4: Table1: please mention unit for each trait. Unit for milk is litre or Kg?

Response 4: Thank you for the observation. The units for all traits in Table 1 are kilograms (kg), as originally indicated in the column header “Trait (kg)”. However, to avoid any ambiguity, we have revised the table to show the unit next to each trait name (e.g., Milk yield (kg), Protein (kg)), as this provides greater clarity for the reader.

Comments 5: Table1: Mention please ow h2 is calculated? By what method? Number of Data? Your model was uni-variate or multi variate?

Response 5: Thank you for this valuable comment. We have clarified in the methodology section (Section 2.3.1, Pedigree-focused best linear unbiased prediction) that heritability was estimated using the average information restricted maximum likelihood (AIREML) method implemented through AIREMLF90, based on a univariate repeatability animal model. The heritability values are derived from the same dataset used in this study, and the total number of records used in the estimation is now specified in the methods section. We have also indicated in Section 2.3.1 that the resulting heritability estimates are reported in Table 1.

Comments 6: L190; Verify please 0.50 or 0.05?

Response 6: Thank you for pointing this out. We have verified the value, and 0.50 is correct as reported in the manuscript. This blending value is commonly used in genomic evaluations and is recommended by various authors, including Lourenco et al. (2020) [DOI: 10.3390/genes11070790].

Comments 7: Tbale2; why accuracy of ABLUP is very low for all traits? In animal breeding, blup animal model for milk traits has suitable accuracy in comparison with traditional models such as Sire model. Please provide possible reasons for very low accuracy.

Response 7: Thank you for your comment. This concern was also raised by other reviewers during the first round. Possible reasons for the low accuracy of the ABLUP model are discussed in the first paragraph of the Discussion section (Lines 302–319). Briefly, the low realized accuracy is mainly due to limited and potentially inaccurate parentage information in the pedigree records, which reduces the reliability of pedigree-based predictions. Similar low ABLUP accuracies have been reported in the literature, especially in datasets with such limitations. Our results also show that including genomic information using ssGBLUP improves accuracy by capturing relationships beyond pedigree records. Furthermore, pedigree completeness among genotyped animals in this study was generally high, indicating that factors other than pedigree gaps influenced the observed accuracy. We have ensured this explanation is clearly presented in the revised manuscript.

Comments 8: L428-429: please provide some justification for this. How you arrived to this conclusion that relatedness or model characteristics influence accuracy of ABLUP? Provide some documents or calculated parameters of your analysis.

Response 8: Thank you for your comment. We acknowledge the importance of avoiding unsupported conclusions. In the revised conclusion, we have clarified that relatedness and model characteristics are likely contributors to the observed low ABLUP accuracy, rather than definitive causes. This interpretation is supported by the discussion in Lines 308–319, where we show that individual accuracies vary substantially and are influenced by the composition of the reference population and pedigree depth. Additionally, Table S1 provides evidence of variability in individual prediction accuracies, which can be linked to factors such as relatedness and connectedness.

Comments 9: L438: Why is the emphasis on the limited number of genotyped animals? Is the effect of the number calculated or simulated? Or is it guessed? Or is it stated based on the bibliography of the subject?

Response 9: We appreciate this question. The emphasis on the limited number of genotyped animals is based on well-established findings in genomic prediction literature. It is not a speculative statement but rather grounded in both theoretical foundations and empirical studies.

The importance of the size of the reference population—particularly the number of animals with both genotypes and phenotypes—has been consistently highlighted since the foundational work by Meuwissen et al. (2001), who showed that prediction accuracy is strongly influenced by the size and quality of the reference population. Increasing the number of genotyped animals enhances the estimation of marker effects and improves the accuracy of genomic predictions. This principle has since been validated in multiple studies (e.g., Clark et al., 2012; Van Grevenhof et al., 2012).

Furthermore, the introduction of strategies such as genotyping females, using multi-breed reference populations, and incorporating historical or imputed data have all aimed to increase the reference population size precisely because of its impact on accuracy.

In our case, while we did not conduct a formal simulation to isolate the effect of reference population size, the consistently low accuracies observed in models using limited genotypic data align with these established findings.  

4. Response to Comments on the Quality of English Language

Point 2: The English is fine and does not require any improvement.

Response 1: Thank you for the positive feedback. We have reviewed the manuscript again and made minor language edits where necessary to improve clarity and consistency, while retaining the overall tone and flow.

5. Additional clarifications

Round 2

Reviewer 3 Report (Previous Reviewer 3)

Comments and Suggestions for Authors

Dear Author,
To my knowledge, for research on genomic selection in Holstein cattle—particularly concerning the optimization of the H matrix within genomic selection models—the dataset used in this study is relatively limited. Substantially expanding the data size would be necessary to enhance the representativeness and reliability of the results. These comments reflect my personal perspective.

This manuscript is a resubmission of an earlier submission. The following is a list of the peer review reports and author responses from that submission.

Round 1

Reviewer 1 Report

Comments and Suggestions for Authors

Comments

  1. Line no. 84 to 87, Objective Clarity:
    In the statement of the study's objective: "to evaluate the accuracy of EBVs for South African Holstein cattle without phenotype records..."the phrase is confusing.

It suggests that only animals without phenotypes were evaluated, rather than focusing on the predictive performance for such animals.

Additionally, the term "EBV" should be corrected to "GEBV," as the predictions incorporate genomic information.

  1. Section 2.1, line no 96 and 112 to 113, Lactation Records Clarification:

There is inconsistency regarding lactation records. One section mentions using the first three lactations from 1989 to 2016, while another states only first-lactation records were considered. Please clarify and ensure consistency throughout.

  1. Section 2.3.2, line no. 138, Variance Component Estimation:

The manuscript states that an ABLUP model was used to estimate variance components. Please specify which approach was used for this analysis (e.g., REML?).

  1. Section 2.3.2.1. Blending

 Blending Coefficient Range: It is noted that an appropriate blending weight typically ranges from 0.80 to 0.99 to ensure that the genomic relationship matrix (G) is positive definite, as per the BLUPF90 guidelines. Please clarify why a blending value as low as 0.6 was selected in this study, as this seems inconsistent with common practice.

  1. Section 3.1 Table 2. Low ABLUP Accuracy:
    The reported ABLUP accuracy (0.01) is extremely low, which is not acceptable. A justification for this low value should be provided.
  2. Results Presentation:

Including graphical representations (such as plots of accuracy vs. parameter values) would make the trends much easier to understand and enhance the clarity of the findings

  1. Interpretation of Blending Results (Lines 224–234, 295–317):

The discussion around the blending coefficient β could be expanded. While β = 0.95 is often used as a standard, the claim that lower β values improve accuracy contradicts findings from other studies. It would be valuable to clarify the conditions under which a lower β may be advantageous and how factors such as population structure and sample size might influence this outcome.

  1. The role tuning and roles of τ and ω in scaling the G and A22 matrices are not clearly explained in the main text. Give brief intuitive explanation how these parameters influence the compatibility of the matrices.
  2. Grammar and Wording Improvements:
  • Line 22: Change “shows the importance” to “show the importance” for subject-verb agreement.
  • Line 20: Consider rephrasing “not tuning produced slightly better results” to better convey that default settings (no tuning) occasionally led to optimal performance.
  • Line 319 to 320: Instead of "tuning is necessary to ensure...", consider rephrasing to:

"Tuning ensures compatibility between G and A22 matrices."

  • Line 350: The phrase "the potential benefits of higher τ values were not fully explored" could be made more assertive, e.g.:"Further exploration of higher τ values could reveal additional benefits.

Reviewer 2 Report

Comments and Suggestions for Authors

There are a few comments and suggestions needed to explained in the manuscript.

P2, L74-76. Please add more backgroud with why blending and tuning value has strong impact on evaluation. BLUPF90 software, by default, already uses optimum values.

P3, Table1. How the authors obtained the VCE. Do you add SNP information in the analysis. SIngle trait or multiple trait?

P4, Section 2.3. The author should add section for variance component estimation. It is unclear if you add SNP information in VCE. What is VCE method you used? SIngle trait or multiple trait was used in VCE and ssGBLUP this study.

P4, L142. From the model equation, please explain why the authors used single trait analysis instead of multiple trait analysis?

P5, Section 2.3.3. Cross validation needed more details. How number of 390 for genotyped cows come from? How many sampling has been done. Please give a formula for realized accuracy.

P5, L202. Does this mean data with all phenotypes, all animal in pedigree, and 1,473 genotyped animals?

P5, L203. Does this mean the daset includes only phenotyped from 390 genotyped data?

P6, L211. The result from this table is questionable. Changing the accuracy up to 30% is too high to be true that come from the different weight without bias. Please check carefully. How you know these weights did not increase the accuracy with upward bias.  2) Does this weights effect the ranking of bulls and dams?

P7, L253. In practice, we never use tau and omega independently. It is okay only for research.

Reviewer 3 Report

Comments and Suggestions for Authors

This study investigates the parameter optimization of the H matrix in the SSGBLUP method, which holds significant research value. However, the dataset employed in the article is relatively limited (with only 1,221 genotyped individuals retained after quality control), and it is recommended to supplement the experimental data with a larger sample size.